# In *Leishmania major*, the Homolog of the Oncogene PES1 May Play a Critical Role in Parasite Infectivity

**DOI:** 10.3390/ijms222212592

**Published:** 2021-11-22

**Authors:** Miriam Algarabel, Celia Fernández-Rubio, Katerina Musilova, José Peña-Guerrero, Andrés Vacas, Esther Larrea, Paul A. Nguewa

**Affiliations:** 1Department of Microbiology and Parasitology, ISTUN Instituto de Salud Tropical, IdiSNA, Instituto de Investigación Sanitaria de Navarra, Universidad de Navarra, E-31008 Pamplona, Spain; malgarabel@alumni.unav.es (M.A.); cfdezrubio@unav.es (C.F.-R.); kmusilova@alumni.unav.es (K.M.); jpena.1@alumni.unav.es (J.P.-G.); avacas@alumni.unav.es (A.V.); 2ISTUN Instituto de Salud Tropical, IdiSNA, Instituto de Investigación Sanitaria de Navarra, Universidad de Navarra, E-31008 Pamplona, Spain; elarrea@unav.es

**Keywords:** *Leishmania*, pescadillo, PES, ribosome biogenesis, infection

## Abstract

Leishmaniasis is a neglected tropical disease caused by *Leishmania* spp. The improvement of existing treatments and the discovery of new drugs remain ones of the major goals in control and eradication of this disease. From the parasite genome, we have identified the homologue of the human oncogene *PES1* in *Leishmania major* (*LmjPES*). It has been demonstrated that PES1 is involved in several processes such as ribosome biogenesis, cell proliferation and genetic transcription. Our phylogenetic studies showed that *LmjPES* encodes a highly conserved protein containing three main domains: PES N-terminus (shared with proteins involved in ribosomal biogenesis), BRCT (found in proteins related to DNA repair processes) and MAEBL-type domain (C-terminus, related to erythrocyte invasion in apicomplexan). This gene showed its highest expression level in metacyclic promastigotes, the infective forms; by fluorescence microscopy assay, we demonstrated the nuclear localization of LmjPES protein. After generating mutant parasites overexpressing *LmjPES*, we observed that these clones displayed a dramatic increase in the ratio of cell infection within macrophages. Furthermore, BALB/c mice infected with these transgenic parasites exhibited higher footpad inflammation compared to those inoculated with non-overexpressing parasites.

## 1. Introduction

Estimates suggest that approximately one tenth of the world’s population lives in extreme poverty (USD <1.90/day), mostly affected by Neglected Tropical Diseases (NTDs) [1]. Disability Adjusted Life Years (DALYs) related to NTDs are constituted for 56% by Years Lost due to Disability (YLD) and for 44% by Years of Life Lost (YLL) [2]. Some NTDs that cause significant mortality, such as leishmaniasis and trypanosomiasis, require medical supervision for their control and depend on expensive and toxic drugs with long courses of treatment [1]. For most NTDs, the development of new drugs is predominantly or exclusively supported by “charitable” funding from public or philanthropic organizations, with little or no contribution from the pharmaceutical industry [3]. The World Health Organization (WHO) recognizes up to 20 diverse illnesses as NTDs, including trypanosomatid-caused pathologies. Among them, leishmaniasis is a group of vector-borne diseases produced by protozoan parasites from the *Leishmania* genus. Over 90 sandfly species are known to transmit *Leishmania* parasites. Clinical manifestations range from self-healing cutaneous lesions to visceral forms which can cause death if untreated. According to the Global Health Observatory, out of 200 territories reporting to WHO in 2020, 98 were endemic for leishmaniasis: 71 countries are endemic for both visceral leishmaniasis (VL) and cutaneous leishmaniasis (CL), 8 for VL only, and 19 countries are endemic for CL only [4]. The treatment of leishmaniasis depends on several factors including immune status of the host, parasite specie and geographic location. In general, healing requires an immunocompetent system [5]. First-line treatments comprise pentavalent antimonials. However, their effectiveness has dramatically decreased due to the appearance of resistances. Alternative chemotherapy such as Paromomycin, Amphotericin B, Pentamidine and Allopurinol showed similar problems [6]. Currently, the only available oral treatment is Miltefosine, for which resistances have also been reported [7,8]. Since leishmaniasis is considered as an NTD, in spite of the large population suffering it the funds for the development of treatment options remain limited; thus, the development of newer therapeutic options is an urgent need. Consequently, research in the field of drug development is directed towards the study of target molecules for the control of this disease [9]. The availability of the sequenced genome from several species of *Leishmania* has allowed the identification of the *Pescadillo* (*PES*) homologue. This oncogene is conserved among eukaryotes and has been mainly related to ribosomal biogenesis as part of the PeBoW complex [10,11,12]. Besides this process, PES involvement has been demonstrated in other important biological routes, including cell proliferation [13,14], embryo development [15] and gene transcription [16]. In addition, high levels of *PES* have been detected in several tumours such as colon, breast, or prostate cancer [17,18,19,20]. On the other hand, the *PES* homologue detected in the fungal pathogen *Candida albicans* was demonstrated to be an essential switch for the yeast-inducing stage, which is responsible for fungal spread throughout the body [21]. In relation to the virulence process, an orthologue of PES has been reported in *Plasmodium falciparum*, exhibiting a protein expression level dependent on parasite stage [22]. Therefore, PES participates in several biological events and is mainly considered as part of the protein set in charge of ribosome biogenesis, a process that takes place in the nucleolus. The synthesis of ribosomes is one of the most energetically demanding cellular processes and involves highly coordinated steps, including the synthesis of ribosomal proteins (RPs) in the cytoplasm, the synthesis and modification of ribosomal RNAs, the importation of rRNAs into the nucleus, the assembly of RPs and rRNAs in the nucleoplasm and the transportation of the two mature subunits (40S and 60S) into the cytoplasm [23,24]. Some findings have shown that the nucleolus is also implicated in biological events, including the cellular stress response [25]. The structure and function of the nucleolus have been mainly analysed in vertebrates and yeast; the knowledge of this nuclear body in trypanosomatid protozoan parasites still needs to be explored [26]. Since *Leishmania* spp. are heteroxenous parasites, they display several development stages and morphological changes, regulated by differential gene expression, in response to the different stress environments present in each host [27,28]. For instance, it has been observed that nutritional starvation found during the metacyclogenesis process can induce the dispersion of nucleolar material into the parasite nucleoplasm [29]. Our work was focused on the discovery of the homologue of *PES* in *Leishmania major*, and we also attempted to shed some light on its involvement in parasite biology and its role as a therapeutic target against pathogenic protozoan.

## 2. Results

### 2.1. LmjPES Is Highly Conserved among Trypanosomatid Cluster

*LmjPES (LmjF.04.0810)* sequence was retrieved from gene DB as a protein-coding gene in *Leishmania major*. This gene is located on chromosome 4 in all the *Leishmania* species and is highly conserved in trypanosomatids. It is found on chromosome 9 and 8 in *T. brucei* and *T. cruzi*, respectively [30,31]. On the other hand, the identity analysis showed that *L. major* gene-sequence identity was higher than 85% and 63% when compared to *L. mexicana* or *L. braziliensis*, and *L. donovani* or *L. infantum*, respectively. In addition, such an identity was also higher than 60% when compared to *Trypanosoma* spp. genes. *LmjPES* encodes a protein of 671 amino acids with an estimated molecular mass of 77.29 kDa and a theoretical pI of 9.69. This protein was predicted to contain two main conserved domains: an N-terminal pescadillo-like protein (NPLP) domain and a BRCT domain (Figure 1A) as found through their secondary structure analysis. To our knowledge, LmjPES is the first reported protein harboring a BRCT-domain in *Leishmania* spp. Protein sequence alignment showed more than 90% of similarity among *Leishmania* species, with *L. infantum* and *L. donovani* the most divergent species (Appendix A). These alignments revealed that PES homologue is a conserved protein in the trypanosomatids cluster (Figure 1B), since *Trypanosoma* spp. conserved 82% of their sequence identity when compared to *Leishmania* spp. The sequence analysis allowed confirmation of the aforementioned NPLP and BRCT domains (Figure 1A). On the other hand, no putative myristoylation signal peptide or Glycosylphosphatidilinositol (GPI)-anchor was detected within the LmjPES sequence. No transmembrane helix was found along the protein. Regarding protein–protein motifs, two coiled regions were identified, covering 60 and 20 amino acids each (Figure 1A). A disordered region was initially described in the LmjPES C-terminal region (residues 405–671); however, a gap junction-related Neuromodulin_N-domain (MAEBL) (residues 499–665) was also recently located at C-terminus. Furthermore, there are two nuclear localization signals (NLS) at the C-terminal region, along the MAEBL-domain (Figure 1A). Gene Ontology analysis (GO terms) mainly showed LmjPES as a putative ribosomal protein as well as an RNA binding protein, among others molecular functions. Regarding biological processes, GO terms predicted LmjPES would be involved in cell proliferation, RNA processing, and the ribosomal subunit biogenesis and cell cycle. Furthermore, the phylogenetic analysis revealed that PES orthologues are not restricted to kinetoplastid organisms. The alignment of PES orthologous sequences showed high conservation levels among eukaryotic organisms. PES orthologues from trypanosomatids and from non-kinetoplastid organisms (*Xaenopus laevis, Dario rerio, Homo sapiens, Mus musculus, Plasmodium falciparum and Saccharomyces cerevisiae*) were used to reconstruct the PES phylogram (Figure 1B). Consistent with our study, PES from *Leishmania* species belongs to a tight and conserved clade where PES orthologs from the other Kinetoplastea species were included. Interestingly, the phylogenetic tree showed that PES orthologue proteins from non-kinetoplastid organisms could be clustered externally to the tightly organized kinetoplastid branches. *H. sapiens* PES1 and *S. cerevisiae* Nop7 belong to the same branch, in which *X. laevis* and *D. rerio* PES share the same cluster (Figure 1B). Based on the homology observed amongst PES sequences from trypanosomatids, we selected *Leishmania major* (LmjPES) to perform further experiments.

### 2.2. LmjPES Is a Protein Located in the Nucleus

The secondary structure analysis of PES homologue in *Leishmania* predicted the presence of NLS, and therefore, a nuclear location. To validate such a prediction, the plasmid pXG-mCherry34-*LmjPES* (Figure 2B) was generated through the N-terminal fusion of *LmjPES* to the red fluorescent fusion protein mCherry34 (Figure 2A). *LmjPES* sequence was obtained from *L. major* cDNA using the primers LmjPESmCherryFw and LmjPESmCherryRv. The subsequent nucleotide sequence was compared to the TriTrypDB *Lmj.04.0810* annotated entry by sequencing. More than 99% sequence identity was found, and no genetic variations were identified between both sequences. Despite the two differences being detected at positions 178 and 550, these double-nucleotide polymorphisms did not change the translated protein sequence (Appendix A).

Fluorescent parasites harbouring pXG-mCherry34 and pXG-mCherry34-*LmjPES* constructs are depicted in Figure 2. *Leishmania* parasites with pXG-mCherry34, used as a control, can be seen in the top lane while parasites with pXG-mCherry34-*LmjPES* are present in the bottom lane (Figure 2C). As observed in the control parasites, red fluorescence did not show specific organelle localization, whereas in cells transfected with pXG-mCherry34-Lmj*PES* plasmid, LmjPES fusion protein can clearly be seen within the nucleus (Figure 2C).

### 2.3. LmjPES Gene Expression was Highly Expressed in the Metacyclic Stage

Having described the presence of the *PES* homologue in *Leishmania* parasites, we aimed to study the expression level of this gene along the promastigote life cycle. For this purpose, qPCR analysis was performed and PES homologue (*LmjPES*) expression was quantified in both procyclic and metacyclic promastigotes. As shown in Figure 3, *LmjPES* metacyclic forms selected by PNA method exhibited the highest expression levels of this gene, more than eight times higher (8.14 ± 0.11) than those observed in procyclic ones. The promastigotes negative-selected by PNA method corresponded to the parasite stage, which is able to infect immune cells. Therefore, *LmjPES* expression might be related to infectivity.

### 2.4. LmjPES Overexpression Increased the Infectivity Rates In Vitro

Since *LmjPES* reached its highest gene expression level in the metacyclic stage, we decided to elucidate the role of this gene in the parasites’ acquisition of infectivity and in the progression of the disease by evaluating cells harbouring episomally overexpressed *LmjPES*. The episomal expression vector pXG-HygR-*LmjPES* was constructed by the strategy previously described (Figure 4A), and used to overexpress *LmjPES* in *L. major* parasites (Figure 4B). The qPCR gene expression measurement of *LmjPES* in the developed transgenic *L. major* strain allowed us to select two clones (pXG-HygR-*LmjPES* 1 and pXG-HygR-*LmjPES* 2, also named clone 1 and clone 2, respectively) which exhibited similar rates of overexpression (6.36 ± 0.24 and 7.62 ± 0.10, respectively) (Figure 4C). Growth curves and cytometry analyses demonstrated the *LmjPES*-overexpressing parasites displayed a growth curve and cell cycle profile similar to control parasites (data not shown). 

Peritoneal macrophages obtained from BALB/c mice were infected with Control (harboring pXG-HygR plasmid) and *LmjPES*-overexpressing metacyclic promastigotes at a 20:1 (promastigotes:macrophage) ratio. Three hours post-infection, significant differences were observed in the percentage of infected cells (20.86 ± 3.42 for control vs. 34.16 ± 3.81 in clone 1 and 39.47 ± 5.67 in clone 2) (Figure 5A) and in the number of amastigotes per infected cell between Control and overexpressing parasites (control: 78.33 ± 5.86 vs. clone 1: 160.00 ± 2.83 and clone 2: 212.00 ± 4.24) (Figure 5B). In fact, the highest infection matched clone 2 (pXG-HygR-*LmjPES2*), which showed the highest level of *LmjPES* gene expression. Six hours post-infection, the percentage of infected macrophages as well as the number of intracellular amastigotes remained higher in *LmjPES*-overexpressing strain infections (30.50 ± 5.70 vs. 85.04 ± 3.47 and 96.50 ± 0.71 vs. 421.00 ± 14.14, respectively) (Figure 5A,B). Figure 5C illustrates the nuclei of macrophages and intracellular amastigotes that were stained with Giemsa to quantify the infection by microscopy analysis.

### 2.5. LmjPES Overexpressing Parasites Exhibited Higher and Faster Footpad Inflammation in BALB/c Mice than Non-Overexpressing Parasites

To evaluate the role of *LmjPES* in virulence *in vivo*, the integrative plasmid pLEXSY-HygR was used to overexpress the *PES* homologue in *L. major* parasites. *LmjPES* sequence was obtained from *L. major* cDNA with both primers, LmjPESpLEXSYFw and LmjPESpLEXSYRv, and then inserted into pLEXSY-HygR plasmid (Figure 6A-B) as described in the *Material and methods* section. Following the strategy previously described, the sequence target gene was integrated by homologous recombination into the chromosomal 18SrRNA (ssu) locus (Figure 6A) and then transcribed by RNA polymerase I. *LmjPES* gene expression levels were measured by qPCR. One clone overexpressing ten times more (10.36 ± 0.06) than control (cells harbouring pLEXSY-HygR plasmid) was selected for *in vivo* infections (Figure 6C). 

Female BALB/c mice were subcutaneously inoculated in the right footpad with three low-dose inoculations of PNA selected metacyclic promastigotes once a week, and the inflammation was monitored weekly (Figure 7A) until the end of the experiment. A slight swelling was detectable in infected animals two weeks after the last inoculation. As illustrated in Figure 7B-C, overexpressing parasites caused a larger area of inflammation compared to control. Four weeks after the last inoculation, the footpad inflammation was higher in animals infected with the overexpressing strains, and such differences were statistically significant by the end of the assay (Figure 7A–D). To assess whether the overexpression of *LmjPES* would induce an increase in lesion size (>0.60 mm), we carried out an additional study. Lesions were allowed to grow until day 35 (five weeks) after the last inoculation. It was observed that the percentage of animals displaying a significant lesion size (>0.60 mm) was 0%, 47.62%, and 65.63%, for the uninfected, positive controls (pLEXSY-HygR), and pLEXSY-HygR-*LmjPES* infected mice, respectively (Figure 7E). The curves were statistically different when compared to that of the pLEXSY-HygR-*LmjPES* group (*** *p* < 0.001), as assessed by log-rank (Mantel–Cox) analysis (Figure 7E). The median time (MT) to overcome a lesion size of 0.60 mm was four weeks in the pLEXSY-HygR-*LmjPES* group (MT= 4 weeks, lesion size >0.60 mm in 100% of the animals), whereas such lesion sizes in the uninfected and pLEXSY-HygR groups remained almost unchanged (lesion size was <0.60 mm in more than 60% of mice after five weeks) (Figure 7E). Thus, an increased expression of *LmjPES* in *L. major*-infected mice correlated with a significantly higher lesion size.

### 2.6. In Vivo Infection with LmjPES Overexpressing Parasites Induced a Higher Production of iNOS in the Inoculation Area

The staining of hematoxylin-eosin footpad sections was also analyzed. It showed the general layout and distribution of cells and provided an overview of tissue sample structure. A higher infiltrate area was observed in mice infected with *LmjPES*-overexpressing parasites compared to those inoculated with pLEXSY-HygR-harboring *Leishmania* (Figure 8A-B). In addition, an immunohistochemical analysis of iNOS protein in the footpad section of mice was performed at the end of the experiment. As observed in Figure 8, five weeks after the last inoculation, iNOS staining was significantly higher in footpad sections from mice infected with *LmjPES* overexpressing parasites compared to the levels of protein in animals inoculated with control (cells harboring pLEXSY-HygR plasmid) (Figure 8B).

## 3. Discussion

The *Leishmania* genus comprises heteroxenous parasites which acquire the capability to infect vertebrates through morphological and molecular changes. In the present work, the *pescadillo* homolog from *Leishmania* parasites has been firstly characterized and related with parasite *in vitro* and *in vivo* infective processes.

The *Pescadillo* (*PES*) gene was first discovered as a site of a retrovirus-insertion mutation that caused severe defects during embryogenesis in Zebrafish [32]. This gene has also been related to several cell processes, such as embryo development, cell proliferation, ribosome biogenesis, cancer and virulence acquisition [14,15,16,18,21,33,34,35]. Interestingly, based on its sequence and function among species, *PES* product is a well-conserved protein. Moreover, its homologues have been identified and functionally well characterized in the human (PES1), yeast (YPH1, Nop7p), and mouse (Pes1) proteomes [36,37]. However, this protein has never been studied or analyzed within trypamosomatids, including *Leishmania* parasites. The genome availability of several *Leishmania* species allowed us to detect the presence of *PES* homolog in *L. major* (*LmjPES*). Due to its high conservation and the importance of the cellular processes in which PES protein is involved, we aimed to study its role in *Leishmania* parasites. The sequence analysis of the conserved motifs and prediction based on alignments showed that LmjPES protein has an evolutionary highly conserved N-terminal domain of pescadillo-like proteins (NPLP-domain) composed of 260 amino acids. These domains have traditionally been related to proteins involved in ribosome biogenesis [38,39]. In fact, previous studies have demonstrated the role of the NPLP domain in PES1 incorporation to PeBoW complex through its interaction with Bop1 [38,39]. The PeBoW complex is essential for cell proliferation and maturation of the large ribosomal subunit in mammalian cells [40]. It is composed of PES1, Bop1 and WDR12 proteins or their yeast homologues, Nop7, Erb1 and Ytm1, respectively [39]. Ribosome biogenesis is a fundamental process that is tightly related to cell growth and proliferation. 

NPLP-domain within LmjPES is followed by a BRCT fragment, which belongs to a complex group of protein domains first described in the tumour-suppressor protein BRCA1. BRCT domains are DNA/protein binding modules mostly implicated in cell processes such as DNA Damage Response (DDR), DNA repair, and/or cell cycle control [41] in a wide range of organisms from bacteria to mammals. BRCT from PES protein has been reported as being responsible for PeBow complex assembly. Additionally, in DNA repair and the cell cycle proteins harboring these domains are involved in pathways such as ribosomal RNA processing and protein interaction in a phospho-dependent manner [42,43,44]. These data suggest that PES protein may have functions beyond ribosome biogenesis. Interestingly, a merozoite adhesive erythrocytic binding ligand (MAEBL) was recently identified at the C-terminal. This domain was first reported in rodent malaria parasites as part of a protein family from the apical organelle of merozoites, which is involved in erythrocyte invasion [45]. To our knowledge, it is the first time that this domain has been reported in *Leishmania*. Further research will be needed to determine if the LmjPES MAEBL domain performs a similar function, or if it is just an evolutionary footprint from non-conserved domains. Since this domain belongs to proteins involved in host cell adhesion, and the adhesion process is critical during infectivity, the presence of the aforementioned domain in LmjPES could show that this protein plays a role in the infectivity process within *Leishmania* biology. 

The nuclear localization of LmjPES was initially predicted by the presence of Nuclear Location Signals (NLS) in the sequence of the protein, specifically located in the C-terminal region. We further demonstrated that LmjPES is a nucleus-located protein, supporting previous results showing the nuclear location of PES homologues. 

It is well known that ribosomal proteins can exhibit extra ribosomal functions [46]. We therefore explored additional functions of PES in *Leishmania major*. *LmjPES* gene expression in metacyclic parasites was significantly higher when compared to its level in procyclics. Thus, it might be involved in *Leishmania* infectivity, since it is mainly expressed in the infective forms. Similarly, previous research demonstrated the implication of a PES homolog from *Candida albicans* in both *in vitro* and *in vivo* virulence [21,34]. On the other hand, in *Plasmodium falciparum* a PES ortholog (PfPES) has been identified during the characterization of the first dual phosphatase as a protein, showing significantly increased levels from ring to trophozoite [22]. Although multiple alignments revealed that PfPES contained domains of PES proteins such as BRCT and NLS, our studies illustrated that PfPES and LmjPES are phylogenetically distant. Since it is mainly expressed in the infective forms, LmjPES might be involved in *Leishmania* infectivity. When murine peritoneal macrophages were infected with *LmjPES* overexpressing parasites, we observed increased infectivity ratios in correlation with higher *LmjPES* expression levels after the infection. Our data suggested that this gene might be implicated in the infectious capability of the parasites. *In vivo* analyses showed similar evidences. To induce such overexpression, *L. major* parasites were transfected with a constitutive expression vector harboring *LmjPES* sequence. pLEXSY-HygR-*LmjPES* infected mice showed significantly higher footpad thickness than that of pLEXSY-HygR infected mice. Accordingly, hematoxylin–eosin staining presented a higher infiltration area in footpads from *LmjPES*-overexpressing infected mice relative to control footpads. An increase of the susceptibility to *L. major* generally occurred after the sub-cutaneous (s.c.) inoculation of a relatively high number of parasites into BALB/c mice, and therefore the induction of the early T helper 2 (Th2) response [47,48]. Consequently, infected mice usually succumbed to parasite visceralization. In the present study, mice were inoculated three times in successive weeks with infective metacyclic promastigotes, as previously described [49]; the spleen and lymph nodes were not infected. On the other hand, high iNOS detection corresponding to increased cell infiltration at the site of infection with *Leishmania* has been previously detected [49,50,51,52]. This is likely due to the recruitment of NOS-producing cells to the site of infection during cutaneous leishmaniasis when inflammation occurred [53,54,55,56].

Certain ribosomal proteins from *Leishmania* parasites have been described as antigens detected by the host immune system [57,58]. In addition, *Leishmania* ribosomal proteins combined with CpG oligodeoxynucleotides have demonstrated effectiveness as vaccination against primary [59] and secondary infection [60]. The acquired protection was associated with the induction of an IL-12 dependent specific-IFN-γ [59]. IFNγ and tumour necrosis factor (TNF) acted synergistically to promote the activation of macrophages through the induction of iNOS [61], which was essential to control the burden of *Leishmania* parasites by the production of reactive oxygen species (ROS). In fact, the increased tissue expression of iNOS has been closely associated with resistance to *L. major* [62]. In our study, high iNOS protein production was detected in the footpad of pLEXSY-HygR-*LmjPES*-infected mice five weeks after the last inoculation. This might be due to the putative antigenic nature of PES homologue, since some parasite antigens are conserved intracellular proteins like heat shock proteins, histones or ribosomal proteins [59]. 

Furthermore, as part of PebOW complex, PES has been related to cell proliferation through its involvement in the cell cycle [16]. It has been observed that growth arrest during PES1 depletion was caused by the disruption of rRNA processing, leading to the activation of a nucleolar stress cell cycle checkpoint, or to the depletion of ribosome and the alteration of translation [21]. Similarly, PES1 protein has been demonstrated to play a key role in oncogenic transformation and tumor progression. For instance, *PES1* expression was suppressed in confluent Hela cells, associated with a reduced proliferation, while early passage Hela cells exhibited higher expression levels of *PES1* [36]. Ribosomal proteins are frequently upregulated in tumors and mutations in genes encoding for proteins that regulate rRNA synthesis [63]. Moreover, nucleolar stress is considered a potential anticancer therapy [64], and this phenomenon is produced in trypanosomatids by transcription inhibitors through the agglomeration of RNA binding proteins [65,66]. Interestingly, there are specific drugs which have shown an ability to decrease transcript levels of *PES1* [12] as a mechanism of action. The targeting of PES remains an emerging therapeutic strategy.

## 4. Materials and Methods

### 4.1. Nucleotide and Amino Acid Sequence Alignments

Nucleotide and genomic sequence data from the kinetoplastid homologs of *PES* and the PES protein orthologous sequences were obtained from TriTrypDB [67]. Other kinetoplastid protein sequences not found in TriTrypDB were retrieved through the Basic Local Alignment Search Tool (BLAST) programs using the NCBI reference protein database and EuPathDB35. The *L. major* protein sequence (XP_888589.1) was used as the query sequence for BLAST searches. Multiple sequences alignments were performed using the BLOSUM62 matrix implemented in CLUSTALW program.

### 4.2. Assessment of Secondary Structure and Post-Translational Modifications

The secondary structure overview from LmjPES was predicted using Quick2D, implemented in MPI Bioinformatics Toolkit [68]. LmjPES disordered regions were inspected by the Meta-Disorder (MD) method [69] from the PredictProtein server [70]. The protein function was deduced by InterPro protein sequence analysis and classification [71]. Conserved domains and families were identified using the Pfam database from the European Bioinformatics Institute [72] (Uniprot accession number: 097209). The following tools were used to assess post-translational modifications: the NetNGlyc 1.0 server46 from the Technical University of Denmark was used for N-terminal glycosylation prediction [73]; the MYR Predictor of the IMP Bioinformatics Group assessed PES sequences for possible myristoylation patterns [74]; the TMHMM Server v2.0 from the Technical University of Denmark was used to determine protein transmembrane regions [75]; ProtParam [76] was used to predict the molecular mass of the protein and its theoretical isoelectric point; phosphorylation sites were assessed by neural network prediction using the NetPhos 2.0 server [77], which identifies serine, threonine, and tyrosine phosphorylation sites; the prediction of the signal peptide and cleavage site was performed using the SignalP 4.0 server [78]; the Nuclear Localization Signal (NLS) was predicted by cNLS mapper [79]; and the protein diagram was drawn by using DOG 1.0: Illustrator of Protein Domain Structures [80].

### 4.3. Phylogenetic Analysis

Protein–protein Basic Local Alignment Search Tool (BLASTP) was used to retrieve the ortholog sequences of LmjPES (XP_888589.1) protein. Multiple sequence alignment was performed using MUSCLE v3.8.31 [81,82], as implemented in the European Molecular Biology Laboratory–European Bioinformatics (EMBL–EBI) web server [83]. This alignment was then used to construct a phylogenetic tree by using the Maximum Likelihood method and a JTT matrix-based model [84]. Initial trees for the heuristic search were obtained automatically by applying Neighbor-Join and BioNJ algorithms to a matrix of pairwise distances estimated using the JTT model, and then selecting the topology with superior log likelihood value. The analysis involved 14 amino acid sequences. All positions containing gaps were eliminated, generating a final dataset with a total of 62 positions. The tree was rooted using the sequence of a hypothetical protein, C923_03267 from *Plasmodium falciparum*. Evolutionary analyses were conducted in MEGA X [85].

### 4.4. Parasites and Animals

*L. major* parasites (Lv39c5) were maintained as previously described [86]. For infection experiments and to preserve their infectivity, parasites were obtained from infected BALB/c mouse spleen and maintained in culture with Schneider’s medium for no more than five passages [87]. Animal studies were performed with female BALB/c mice and all the procedures were approved by the Animal Care Ethics Commission of the University of Navarra (approval number: E24/18(068/15) 10 August 2018).

### 4.5. Genetic Manipulation of *Leishmania major*

The plasmids pXG-mCherry 34, pXG-HygR and pLEXSY-HygR were used in this study. Plasmid pXG-mCherry34 was assembled as reported [88]. The coding DNA sequence (CDS) corresponding to *LmjPES* was amplified by PCR from *L. major* genomic DNA with the primers LmjPESmCherry-Fw and LmjPESmCherry-Rv, LmjPESpXG-Fw and LmjPESpXG-Rv, and LmjPESpLEXSY-Fw and LmjPESpLEXSY-Rv (Table 1), for insertion in pXG-mCherry34, pXG-HygR and pLEXSY-HygR, respectively. *LmjPES* PCR products were then ligated into pCR^®^ 2.1-TOPO^®^ (ThermoFisher Scientific, Rockville, MD, USA) cloning vector. Plasmids were digested with the following restriction enzymes: *NotI-HF* (New England Biolabs, Ipswich, MA, USA) for the pXG-*LmjPES*-mCherry34 construct, *BamHI* (Takara, Tokyo, Japan) for the pXG-HygR-*LmjPES* construct, and *NcoI* (New England Biolabs, Ipswich, MA, USA) and *KpnI* (New England Biolabs, Ipswich, MA, USA) for the pLEXSY-HygR-*LmjPES* construct. These digestions were gel purified using QIAquick^®^ Gel Extraction Kit (Qiagen^™^) following the manufacturer’s protocol, and then ligated using T4 Ligase (Invitrogen^™^). *LmjPES* sequences and orientation within pXG-*LmjPES*-mCherry34 and pXG-HygR-*LmjPES* were confirmed by PCR and sequencing. pLEXSY-HygR-*LmjPES* plasmid was linearized by digestion with *SwaI* (New England Biolabs, Ipswich, MA, USA) before electroporation [89]. pLEXSY-INT primers (Table 1) were used to confirm pLEXSY-HygR-*LmjPES* integration in the 18S rRNA locus.

### 4.6. Fluorescence Microscopy

A total number of 2 × 10^7^ log-phase pXG-mCherry34-*LmjPES L. major* promastigotes were fixed using a 1% paraformaldehyde/phosphate-buffered saline (PBS; Gibco Laboratories) solution. Then, cells were stained with a 1 mg/mL DAPI (Sigma Aldrich, St. Louis, MO, USA) solution (30 min at 4 °C) and washed twice with PBS. Images from the slides were acquired in a PerkinElmer ultraVIEW confocal microscope.

### 4.7. In Vitro Infections

Murine peritoneal macrophages extracted as previously described [87] were seeded in 8-well culture chamber slides (Lab- TekTM; BD Biosciences) at a density of 5 × 10^4^ cells per well in Roswell Park Memorial Institute (RPMI) medium and allowed to adhere overnight at 37 °C in a 5% CO_2_ incubator. Metacyclic *L. major* promastigotes harbouring pXG-HygR or pXG-HygR-*LmjPES* plasmids and isolated by the peanut agglutinin (PNA) method [90] were used to infect macrophages at a macrophage/parasite ratio of 1/20, and incubated at 37 °C in a 5% CO_2_ atmosphere. Chamber slides were washed with PBS, fixed with methanol three and six hours post-infection, and finally stained with Giemsa. The number of amastigotes per macrophage was evaluated by light microscopy (200 infected cells were analyzed in each well). 

### 4.8. In Vivo Infections

BALB/c mice were distributed in three groups; mice from one group were PBS inoculated, and two groups were infected with *LmjPES* overexpressing parasites (pLEXSY-HygR-*LmjPES*) and control parasites (pLEXSY-HygR), respectively. The animals were subcutaneously inoculated as reported [49]. Footpad swelling was quantified weekly with a caliper, and the swelling was determined as the difference between the infected and non-infected footpad. Five weeks after the last inoculation, animals were euthanized. For immunohistochemistry determination, footpads were placed in 4% (*w*/*v*) formaldehyde (PanReac, Barcelona, Spain) and then formalin-fixed.

### 4.9. RNA Extraction and Gene Expression Analysis

A stationary phase culture was used to select metacyclic parasites by PNA method as previously described [91]. Total RNA extraction, retrotranscription and qPCR assays were performed as reported [49]. Primers qLmjPES-Fw and qLmjPES-Rv and, primers qGAPDH-Fw and qGAPDH-Rv were used to amplify *LmjPES* and g*lyceraldehyde-3-phosphate dehydrogenase* (*GAPDH*) housekeeping gene [87], respectively, in *Leishmania* parasites (Table 1). The relative quantification of gene expression was calculated by the 2^ct(GAPDH)-ct(gene)^ method [49].

### 4.10. Immunohistochemistry Analysis:

Immunohistochemistry studies were performed as previously described [49], using the primary antibody rabbit anti-iNOS (1:400; Abcam, ab15323). Fiji 2.0 software was used for analysis [91].

### 4.11. Statistical Analyses

Statistical studies were executed with GraphPad Prism v7. The analyses were performed using the non-parametric Kruskal–Wallis and Mann–Whitney U tests. All *p* values were two-tailed and considered significant if *p* < 0.05. Data were represented as mean ± SD. Experimental differences between groups were studied by one-way ANOVA (or by unpaired t test, in the case of two comparisons). Lesion sizes were compared by the log-rank (Mantel–Cox) test. The median time to develop a significant lesion (size > 0.6 mm) was estimated with Kaplan–Meier analysis. *p*-values were considered significant when * *p* < 0.05, ** *p* < 0.01, *** *p* < 0.001 and **** *p* < 0.0001.

## 5. Conclusions

In conclusion, the present work shows the relation between the newly discovered homologue of PES in *Leishmania* parasites with infectivity capability *in vitro* and virulence *in vivo*. On the one hand, transgenic parasites overexpressing *LmjPES* were able to infect a higher percentage of macrophages during *in vitro* infection. On the other hand, *LmjPES* expression in *L. major* parasites produced an increase in cell recruitment at the inoculation site during *in vivo* studies in BALB/c mice. Further studies are needed to better understand the role of this newly discovered gene in leishmaniasis outcomes. However, due to its relevant implications for parasite virulence, LmjPES might be considered a promising target for new drug design against leishmaniasis.

## Figures and Tables

**Figure 1 ijms-22-12592-f001:**
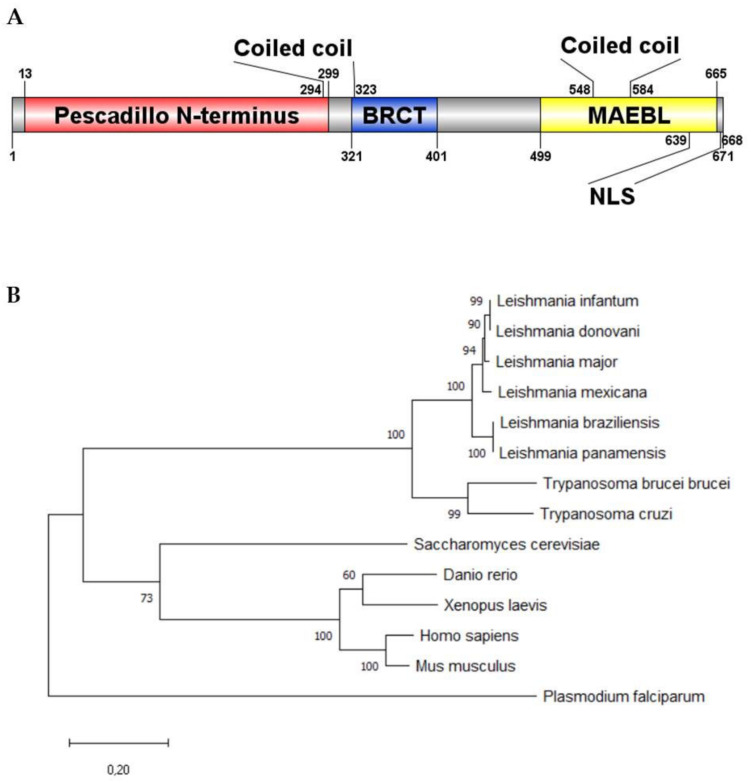
LmjPES is highly conserved among trypanosomatid cluster. (**A**) Schematic representation of the predicted catalytically relevant regions and domains of LmjPES. (**B**) Phylogenetic analysis of LmjPES. Tree was rooted using the sequence of the hypothetical protein C923_03267 from *Plasmodium falciparum*. Protein sequences of PES trypanosomatid orthologues (accession numbers: XP_001561761.1, XP_010703722.1, XP_003871849.1, XP_888589.1, A4HS78.1, AYU75861.1, XP_827157.1, RNF20479.1) and PES proteins form *Danio rerio* (NP_571105.3), *Xenopus laevis* (NP_001080557.1), *Homo sapiens* (NP_055118.1), *Mus musculus* (NP_075027.1) and *Saccharomyces cervisiae* (PJP07134.1) were used to generate the phylogram.

**Figure 2 ijms-22-12592-f002:**
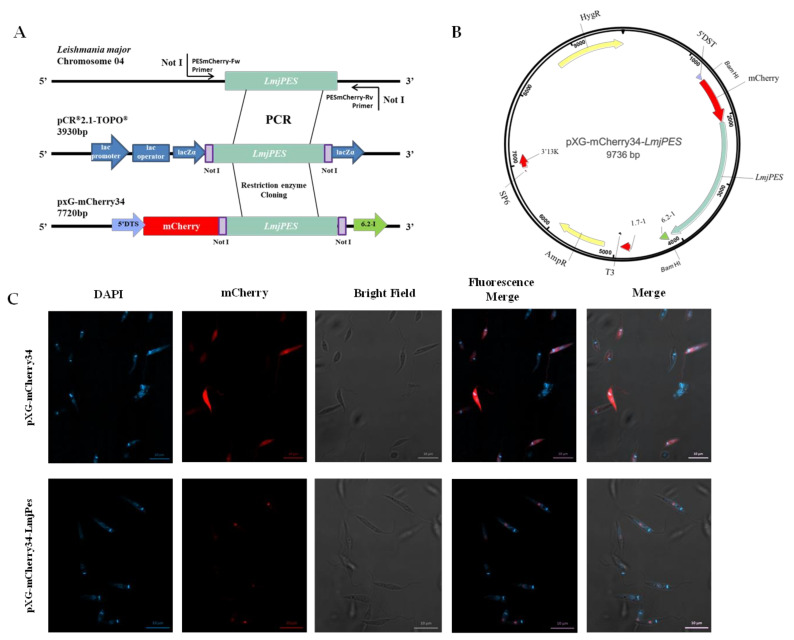
LmjPES is a protein located in the nucleus. (**A**) Schematic representation of the approach to generate the LmjPES red fluorescent protein. (**B**) Map of the plasmid carrying the genetic sequence of the red fluorescent protein mCherry fused to *LmjPES* gene. (**C**) *L. major* promastigotes expressing control mCherry34 (top lane) and mCherry34-*LmjPES* (bottom lane) fusion proteins.

**Figure 3 ijms-22-12592-f003:**
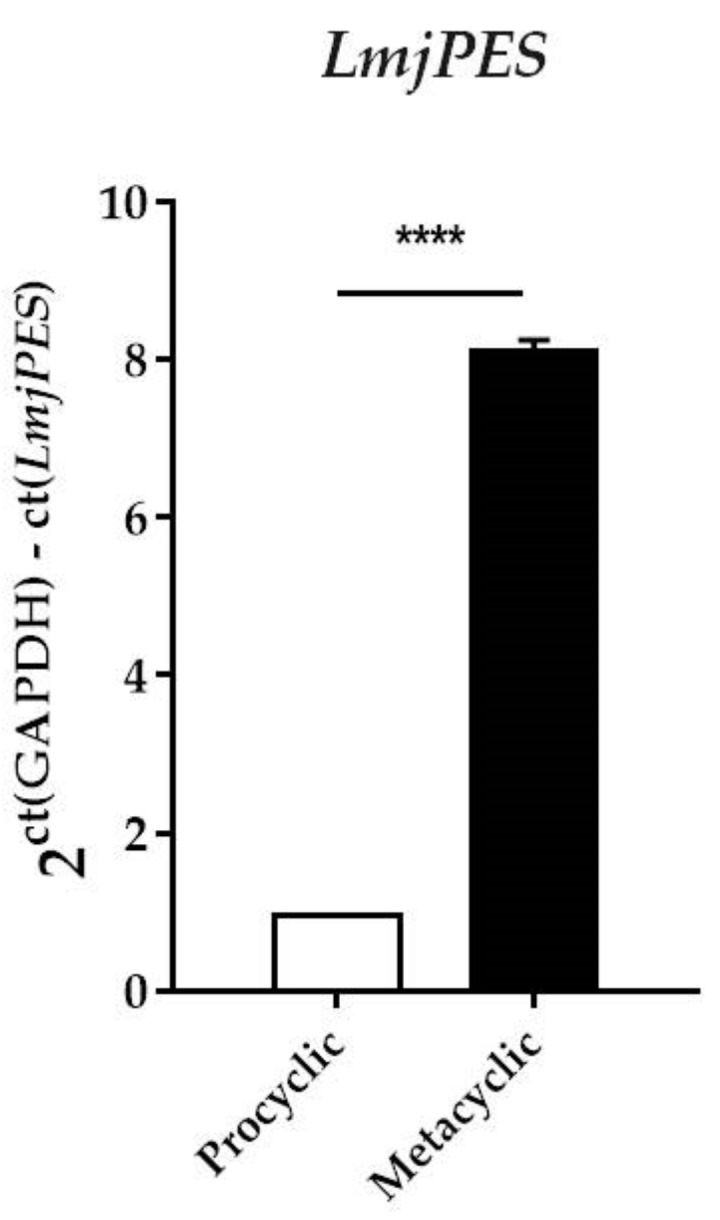
*LmjPES* gene expression is highly expressed in the metacyclic stage. *LmjPES* relative gene expression in procyclic and metacyclic promastigotes from *Leishmania major*. Bars represent gene expression mean fold change (±SD) from three independent experiments (**** *p* < 0.0001).

**Figure 4 ijms-22-12592-f004:**
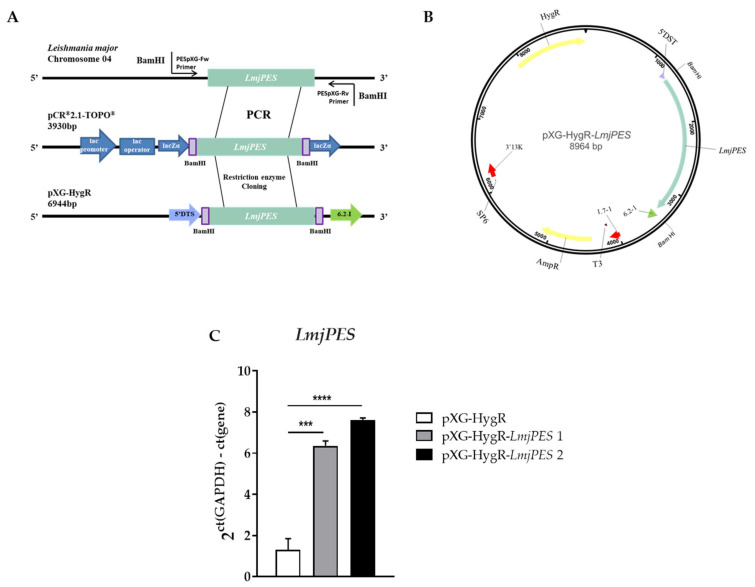
Generation of *Leishmania major* strains overexpressing *LmjPES*. **(A)** Schematic representation of the approach to clone *LmjPES* into the pXG-HygR overexpression plasmid. (**B**) Map of the episomal plasmid carrying the *LmjPES* sequence used to generate overexpressing parasites (**C**) *LmjPES* gene expression level of two *LmjPES*-overexpressing *L. major* strains (pXG-HygR-*LmjPES1* and pXG-HygR-*LmjPES2*) compared to *L. major* control (pXG-HygR) parasites (*** *p* < 0.001; **** *p* < 0.0001).

**Figure 5 ijms-22-12592-f005:**
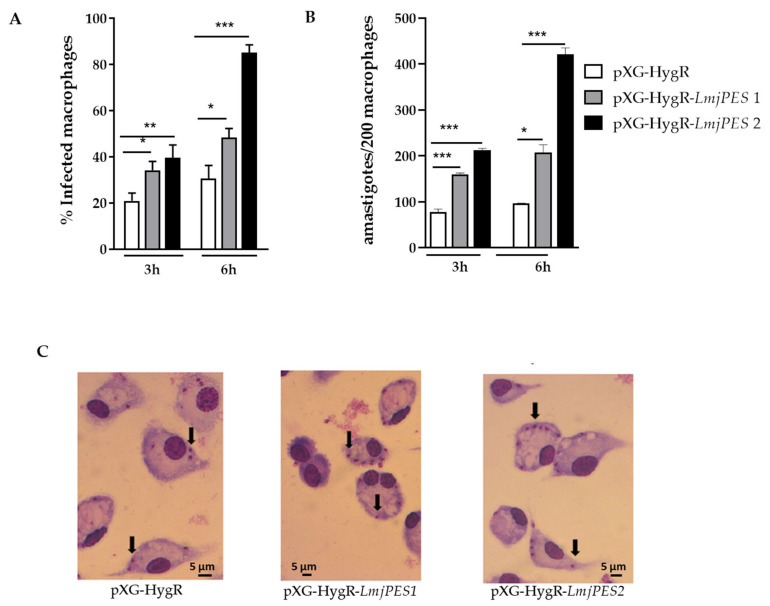
*LmjPES* overexpression increases infectivity rates *in vitro*. Analyses were conducted with two *LmjPES*-overexpressing *L. major* strains (pXG-HygR-*LmjPES1* and pXG-HygR-*LmjPES2*). (**A**) The percentage of infected macrophages and (**B**) the number of amastigotes per 200 infected cells were evaluated by microscopy counting after Giemsa staining at 3 and 6 h post-infection. (**C**) The observed changes in the infectivity rate throughout the *in vitro* infection caused by overexpressing and non-overexpressing parasites were graphed for one representative experiment. Solid black arrows indicate Giemsa-stained amastigotes. Data represent the means (±SD) from the triplicates of at least three independent experiments (* *p* < 0.05, ** *p* < 0.01, *** *p* < 0.001).

**Figure 6 ijms-22-12592-f006:**
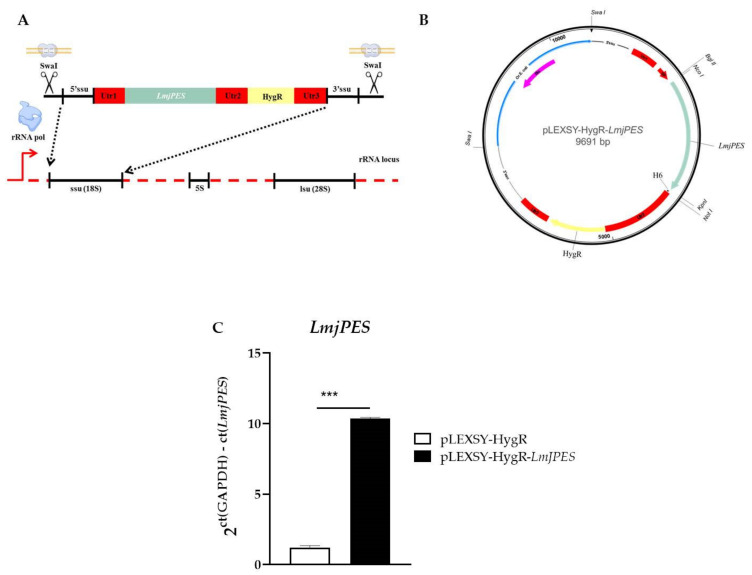
Generation of *Leishmania major* strains overexpressing *LmjPES*. (**A**) Schematic representation of the approach to overexpress *LmjPES* by the integration of pLEXSY-HygR plasmid harboring *LmjPES* sequence (pLEXSY-HygR-*LmjPES*) in the 18S rRNA locus. (**B**) Map of the integrative plasmid carrying the *LmjPES* sequence used to generate overexpressing parasites. (**C**) *LmjPES* gene expression level of transgenic *L. major* strains (pLEXSY-HygR-*LmjPES*) compared to *L. major* control (pLEXSY-HygR) parasites. The *LmjPES* expression level of recombinant parasites was checked by quantitative real-time PCR (qPCR) and represented (*** *p* < 0.001).

**Figure 7 ijms-22-12592-f007:**
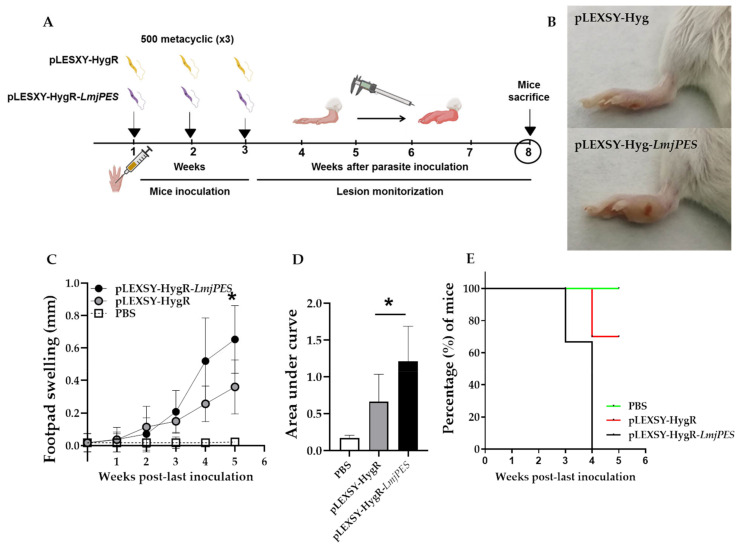
*LmjPES* overexpressing parasites exhibited higher and faster footpad inflammation in BALB/c mice than non-overexpressing parasites. (**A**) Schematic representation of the experimental setting. During the inoculation period, animals were subcutaneously infected in the right footpad with 500 metacyclic parasites, overexpressing (pLEXSY-HygR-*LmjPES*) or non-overexpressing control (pLEXSY-HygR) *LmjPES*. Three cycles of inoculation were carried out once a week. After the last inoculation, the swelling was measured weekly until the end of the experiment (lesion monitorization). (**B**) Representative images of inoculated footpads of mice infected with parasites overexpressing (pLEXSY-HygR-*LmjPES*) or control (pLEXSY-HygR) *LmjPES*. (**C**) Footpad swelling measured weekly until five weeks after the last parasite inoculation with both pLEXSY-HygR and pLEXSY-HygR-*LmjPES* strains. (**D**) Area under the curve for footpad swelling from uninfected (PBS) mice and inoculated with pLEXSY-HygR and pLEXSY-HygR-*LmjPES* parasites. (**E**) Percentage of mice with no significant lesion development after being infected by *LmjPES* overexpressing (pLEXSY-HygR-*LmjPES*) and non-overexpressing parasites (pLEXSY-HygR). Significant lesion was defined as a swelling >0.6 mm. Survival curves were calculated with PRISM version 5.0. The bars represented the means (±SD) (* *p* < 0.05).

**Figure 8 ijms-22-12592-f008:**
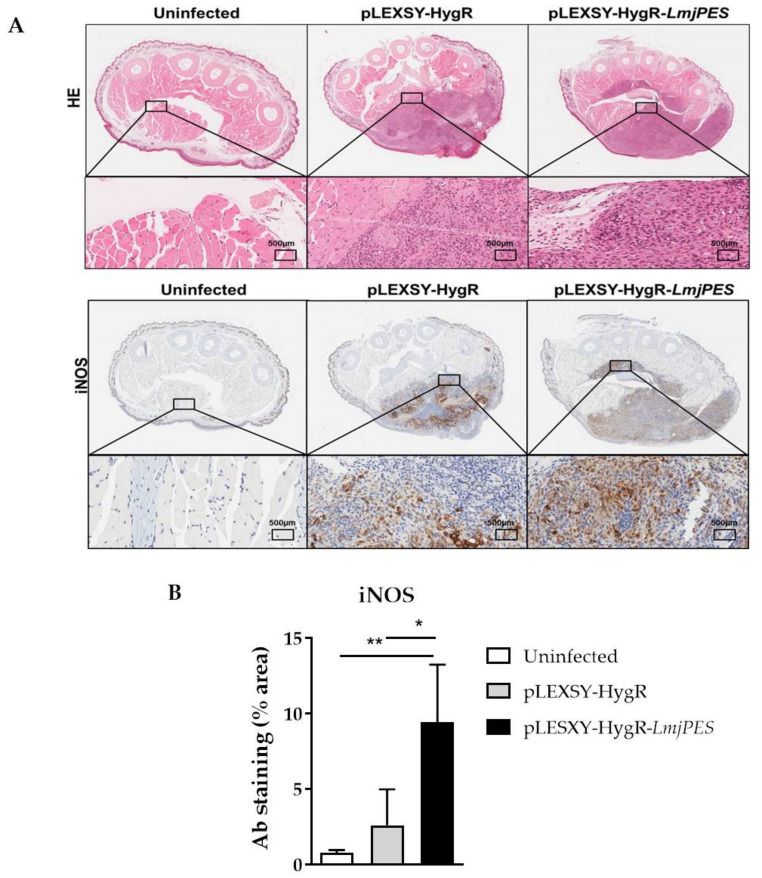
Histopathology and immunohistochemistry results of the footpad sections. Five weeks after last inoculation, animals were sacrificed to determine the footpad inflammation by histology staining methods. (**A**) Hematoxylin-eosin and immunohistochemical staining for iNOS in footpad sections from mice infected with *LmjPES*-overexpressing and non-overexpressing parasites. (**B**) The differences in the area of footpad sections stained with antibodies of iNOS from animals infected with *LmjPES*-overexpressing parasites compared to those inoculated with control (cells harboring pLEXSY-HygR plasmid). The bars represented the means (±SD) (* *p* < 0.05, ** *p* < 0.01).

**Table 1 ijms-22-12592-t001:** Primer sequences employed for the genetic manipulation of *Leishmania* parasites.

Name	Forward (Fw) Primer (5′→3′)	Reverse (Rv) Primer (5′→3′)
LmjPESmCherry	GCGGCCGCTATTATGGTCACATAAGAAGCAG	GCGGCCGCATTACTGCACCCACTT
LmjPESpXG	GCCGGATCCCCACCAATGGTCCATAAGAAGCAGGCA	GCCGGATCCTTACTGCACCCACTTGGGCAG
LmjPESpLEXSY	CCATGGGAAATGGTCCATAAGAAGCAGGCA	GGT ACC CTG CAC CCA CTT GGG CAG TTT
pLEXSY-INT	CCGACTGCAACA AGGTGTAG	CAT CTA TAG AGA AGT ACA CGT AAA AG
qLmjPES	GAGATGGACATGGAGGACGA	TCATCTCGCGCTG
qGAPDH	CATCAAGTGCGTGAAGGCGC	CGTCGGCGAGTACTCGTGCTG

## Data Availability

Not applicable.

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
