# Peer review of "In Leishmania major, the Homolog of the Oncogene PES1 May Play a Critical Role in Parasite Infectivity"

_ijms, 2021, doi:10.3390/ijms222212592_

Round 1

Reviewer 1 Report

The similarity is 26% according to Turnitin, so please reduce it during the revision process. 

  1. Kindly check the abbreviations of this paper.

2. English proofreading and further to align back with the author's instructions of the respective journal. 

Author Response

Thank you.

Reviewer 2 Report

I carefully reviewed the manuscript titled “In Leishmania major, the homolog of the oncogene PES1 may play a critical role in parasite infectivity”. Authors identified for the first time the homologue of the human oncogene PES in Leishmania major and they demonstrated, thought exhaustive experiments, the role of this gene during the infection. The study is well design and executed, methods are clear and the results are clearly reported. The conclusions are fully supported by the results.

The manuscript is good enough for its publication nevertheless, I would suggest few minor changes:

- line 81 “ … … environment present in each host.” I would add a reference to this sentence

-  line 154 “ … pXG-mCherry34, pXG-HygR and pLEXSY-HygR, respectively.”

Only plasmid pXG-mCherry34 have been introduced previously at line 148. Please indicate also the other two plasmids as plasmids used.

- subhead: “2.7. RNA extraction and gene expression analysis” I would suggest authors to change the order.

2.78. In vitro infections:

2.89. In vivo infections:

2.97. RNA extraction and gene expression analysis:

In case authors would prefer to keep their order, please indicate what PNA mean the first time that it is written (line 186) and explain which RNA from mice tissue samples are you extracting (lines 188-189).

- Lines 250-252 “… … brucei and T. cruzi, respectively.” Please add a reference here.

- Figure 2,C. The resolution of Leishmania transfected with pXG-mCherry34-LmjPES plasmid is not very clear. Please adjust the contrast and the resolution of the image.

- Figure S2. The resolution of this image is not good. Please improve it

Author Response

Thank you.
